# Perspectives on the Regional Strategy for Implementation of National Action Plans on Antimicrobial Resistance in the WHO African Region

**DOI:** 10.3390/antibiotics13100943

**Published:** 2024-10-09

**Authors:** Ali Ahmed Yahaya, Walter Fuller, Dennis Kithinji, Yidnekachew Degefaw Mazengiya, Laetitia Gahimbare, Kizito Bishikwabo-Nsarhaza

**Affiliations:** 1WHO Regional Office for Africa, Cité du Djoué, Brazzaville P.O. Box 06, Congo; aliahmedy@who.int (A.A.Y.); mazengiyay@who.int (Y.D.M.); gahimbarel@who.int (L.G.); nsarhazak@who.int (K.B.-N.); 2MedRight Consulting Ltd., Maua P.O. Box 254715149694-60600, Kenya

**Keywords:** antimicrobial resistance, regional AMR strategy, WHO Regional Office for Africa, national action plan, antimicrobial stewardship, one health, AMR awareness campaigns

## Abstract

**Background:** The WHO Regional Office for Africa developed a Member States (MS)-endorsed regional strategy to fast-track the implementation of MS’ national action plans (NAP) on Antimicrobial resistance (AMR). This study explored the perspectives of AMR’s national focal points in MS on the implementation of the priority interventions of the regional strategy in their countries. **Methods:** An online survey consisting of ratings and discussions covering the implementation of the six priority interventions was conducted. Sums of the scores per priority intervention were obtained, and their percentage to the total possible scores were calculated to categorize the implementation as inadequate (0–25%), basic (26–50%), intermediate (51–75%), or advanced (76–100%). **Results:** Thirty-six of the forty-seven national AMR focal points responded to the survey between 12 November 2023 and 8 January 2024. The implementations were rated as 37–62% (basic-to-intermediate), with the multisectoral coordination and collaboration committee receiving the highest overall rating (62%, 421/684), while the promotion of sustainable investment for the NAP on AMR received the least overall rating (37%, 257/700). The focal points mainly recommended awareness campaigns, capacity building, and regulations and guidelines to improve the implementation of the AMR strategy. **Conclusions:** The survey revealed a need to enhance awareness campaigns, support the establishment and functioning of AMR evaluation and monitoring systems, and build the capacity of AMR staff with cost-benefit analysis and budgeting skills. It also showed the necessity to improve awareness and conduct education on AMR, streamline evidence generation through One Health Surveillance systems, integrate initiatives to reduce hospital-acquired infections in the antimicrobial stewardship programs, and enhance regulations and guidelines to optimize the use of antimicrobials.

## 1. Introduction

Antimicrobial resistance (AMR) is a global concern affecting health, livelihoods, food security, ecosystems, and economics. AMR-associated infections cause about five million deaths annually [1]. Nearly 10 million people may die, and economic output cumulatively worth USD 100 trillion may be lost by 2050 due to AMR [2,3]. In 2019, bacterial AMR alone caused approximately 1.3 million deaths [4]. Therefore, AMR is a substantial risk factor for morbidity, disability, and mortality.

AMR-associated morbidities and deaths cause economic and social losses. Livestock productivity is likely to decline by 11% by 2050 with the current trend of AMR [5]. Additionally, the world may lose USD 3.4 trillion of gross domestic product (GDP) annually by 2030, which would be more severe than the economic consequences of the 2008–2009 global financial crisis [6]. An additional 24 million people, mainly in low-income countries, may fall into extreme poverty by 2030 if AMR is not mitigated [6]. Thus, AMR is likely to exacerbate food insecurity and poverty. 

AMR-related deaths are disproportionately high in Africa, thus hampering Africa’s progress toward economic security [7]. Almost 20% of the 1.3 million bacterial resistance-related deaths in 2019 occurred in Sub-Saharan Africa, as reported by Murray et al. [4]. Western Sub-Saharan Africa has the highest AMR-related death rate globally, with 27.3 deaths per 100,000 [4]. Consequently, AMR may worsen the health, economic, and social inequalities and erase the gains made in areas of health, food security, and economic prosperity, thus negatively impacting the achievement of sustainable development goals and other national, regional, and global priorities [6,8]. Prioritizing AMR in the African Region is therefore of utmost importance and urgency to sustain and progress in the areas of health security, health system strengthening, and sustainable development goals (SDGs) by 2030. 

The quadripartite partners on the AMR global action plan (GAP) continue to advocate for the deployment of the One Health approach (OHA) as a central locomotive or a vital currency for building synergies and implementing efficient and impact-oriented AMR mitigation strategies. The effective implementation of global, regional, and national action plans will prove uncertain with the variations in AMR-related initiatives if an OHA that provides sufficient prioritization and eliminates redundancy is not deployed in most member states (MS) [9]. The implementation of an OHA in the African Region has been slow due in part to data challenges, systems-based fragmentation, and poor deployment of the OHA. As a result, only 34% of African countries have implemented NAP in the eight years since the launch of the GAP [10]. 

The WHO Regional Office for Africa, leaning on the lessons in supporting MS to implement their One Health National Action Plans (NAP), appreciated the implementation challenges on the ground and, with demand from MS, developed a Regional AMR strategy over the years. The AMR strategy was endorsed by member states during the Regional Committee’s 73 session held in Gaborone, Republic of Botswana, between 28 August 2023 and 1 September 2023. It aims to leverage the critical MS commitments and build on the investments made in combatting COVID-19, including improved Infection Prevention and Control (IPC) and Water, Sanitation, and Hygiene (WASH) measures, as well as enhanced genomic laboratory capacity to support results-based implementation in the African Region [11,12]. 

The regional AMR strategy is meant to support MS in speeding up the implementation and monitoring of One Health NAPs on AMR. It highlights the importance of ownership and investment by the governments and focuses on avoiding duplication, reducing silos, and eradicating bottlenecks that hamper efforts to address AMR in the African Region [12]. The underlying emphasis on ownership and investment is driven by the fact that less than 2% of countries in the region have a dedicated budget for AMR. Without strong commitment and financial and human capital investment from national governments, the morbidity and mortality in Africa will significantly increase with deleterious social and economic consequences [13,14]. The AMR strategy is, therefore, not just a guiding document but an urgent call to steer the fight against AMR.

The regional AMR strategy aims to foster the sustainable and result-oriented implementation of NAPs on AMR towards reducing AMR-attributable morbidities, disabilities, deaths, and socioeconomic losses [12]. The strategy emphasizes the strengthening of the multisectoral coordination and governance driven by the OHA, improving awareness and understanding of AMR, enhancing knowledge and evidence on AMR rates and antimicrobial consumption, and optimizing the use of antimicrobials in humans [12]. Thus, assessing an MS’s readiness to align with the recommendations in the adopted AMR strategy and prioritize agreed indicators is important to reveal areas that both national and external stakeholders can support. Evidence-based interventions are vital to make the fight against AMR holistic, effective, and sustainable, taking into account other equally important cognate strategies and interventions such as preparedness and response to public health events and primary health care.

This study is an assessment describing the status of AMR strategy-related initiatives in WHO African Region (WHO/AFRO) MS. The survey targeted and obtained information from national AMR focal points (AMR FPs) appointed by WHO as key actors of the NAP implementation in MS. The survey identified priority interventions that need support from the MS to attain the 2030 targets of the AMR strategy. A Comprehensive understanding of the prevailing challenges, weaknesses, threats, and opportunities in the implementation of baseline activities to achieve key milestones and outcomes, as detailed in the AMR regional strategy, is indispensable if 2030 targets are to be achieved. The targets were set by the 47 MS of the WHO/AFRO. This study sought to answer the question: What are the perspectives of AMR FPs on the implementation of “One Health” multisectoral collaboration and coordination, in-country monitoring and evaluation systems, sustainable investment for NAPs, awareness and education, initiatives to reduce the incidence of infections, and optimization of the use of antimicrobials in the fight against AMR?

## 2. Results

Thirty-six of the targeted forty-seven AMR FPs responded to the survey between 12 November 2023 and 8 January 2024. Some of them skipped a few questions, but all of their responses were included in the analysis. The presented results summarize the perspectives of the AMR FPs on the AMR activities related to the regional strategy’s priority interventions in the WHO/AFRO. They highlight activities within the priority interventions that the AMR FPs consider to be lagging behind in the MS and the most appropriate interventions to streamline the implementation of the regional AMR strategy.

### 2.1. Overall Perspectives on the Implementation of Priority Interventions

The AMR FPs viewed the implementation of four priority interventions as intermediate: multisectoral coordination and collaboration committee (MCCC) (62%, 421/684), raising awareness and education (55%, 257/470), building of knowledge and evidence base (51%, 162/315), and reducing infections that predispose to AMR (54%, 474/875). They are regarded as basic in the optimization of the use of antimicrobials (49%, 432/875) and the promotion of sustainable investment for NAP on AMR (37%, 257/700). Overall, the AMR FPs rated the implementation of the AMR strategy-related initiatives 51%, which is intermediate but bordering basic.

### 2.2. AMR Multisectoral Coordination and Collaboration Committee

AMR FPs in Angola, Cameroon, Congo, Gabon, Equatorial Guinea, Mozambique, and Togo viewed that their countries lack an AMR MCCC (Figure 1). MCCCs in Burundi, Benin, and Namibia were identified as non-functional in terms of having defined roles and responsibilities or accountability systems.

Most of the 26 AMR FPs that responded to the question on AMR MCCC were satisfied with the functioning of the AMR MCCC in their countries (Table 1). The functioning of the AMR MCCC in Eswatini was the least satisfying (Figure 2). Most of the member states, except Benin, Burundi, Eswatini, and Niger had structures or monitoring systems for multisectoral One Health collaboration and coordination (Table 1, Figure 2). One Health collaboration activities against AMR in Nigeria, Eswatini, Uganda, Niger, and Eritrea were only a little integrated with other cognate strategies such as universal health coverage and quality improvement. Chad, Eswatini, and Niger had not allocated human and financial resources to One Health’s collaborative activities against AMR (Figure 2).

Twenty-nine AMR FPs responded to the open-ended question on the strategies to strengthen multi-sectoral “One Health” collaboration and coordination on AMR in their respective countries. The AMR FPs mainly suggested awareness campaigns, integrated AMR surveillance, development of diverse guidelines and toolkits, a multisectoral approach to the revision of the NAP, training healthcare professionals, and supporting research and development activities. For example, the AMR FP in Kenya indicated “AMR Awareness campaigns, e.g., World AMR Awareness Week (WAAW)” as the ideal intervention to strengthen multi-sectoral “One Health” collaboration and coordination.

### 2.3. Raising Awareness and Education on AMR

Fifteen of the thirty-six AMR FPs indicated that their countries have communication, education, and awareness campaign strategies on AMR (Figure 3a). The promotion of AMR prevention in the campaign strategy on AMR in Mali has only been implemented a little. Only Chad and Equatorial Guinea were not implementing their campaign strategies on AMR (Figure 2 and Figure 3b). Cameroon, Mozambique, and Togo lacked education and awareness campaign initiatives to target secondary and tertiary learning institutions (Figure 3c). Benin, Burundi, and Nigeria had not attempted to have AMR education in their curricula. Ten AMR FPs expressed that professional bodies and institutions of higher learning are not represented in the AMR MCCCs and AMR technical working groups (TWG) in their countries (Table 1 and Figure 3d). AMR professionals in Zambia, Angola, Comoros, Eritrea, Equatorial Guinea, Mozambique, and Niger were not engaging professional bodies and institutions of higher learning in their countries on AMR (Figure 2). Angola, Comoros, Gabon, Equatorial Guinea, and Niger had not integrated health communication, infectious diseases, and social marketing experts into antimicrobial campaign planning teams.

AMR FPs in 16 MS saw improving allocations in terms of human capacity and financial resources as a key strategy to improving education and awareness of AMR. For example, the AMR FP in Gambia listed “*Increase financial support for AMR coordination*”. The second most prominent theme was the establishment of collaboration frameworks, including diverse players, to drive AMR education and awareness. In this regard, the AMR FP in Madagascar recommended “*Inclusion of the private sector and civil society at both national and sub-national (county) levels*”. The AMR FPs also recommended capacity building, especially among AMR leaders; strengthening legislation, policies, plans, and regulations; and improving intersectoral communication, stakeholder engagement, and routine evaluation to improve education and awareness of AMR.

### 2.4. Building a Knowledge and Evidence Base on AMR

Fourteen WHO/AFRO MS do not submit data to the Global Antimicrobial Resistance and Use Surveillance System (GLASS)-AMR and GLASS antimicrobial consumption (AMC) as per the AMR FPs who participated in the survey (Table 1). Burundi and Mozambique do not entirely generate information on the incidence and prevalence of AMR in various pathogens and diverse geographical areas (Figure 2). Angola and Benin do not use the information on the incidence and prevalence of AMR to guide clinical care and policy actions as well as monitor the effectiveness of interventions. Cameroon and Ghana use the information to guide clinical care and policy actions but not to monitor the effectiveness of interventions (Figure 2).

Twenty-two AMR FPs recommended interventions to build a knowledge and evidence base on AMR in their respective countries. Establishing an information system that relays surveillance data and reports in real-time emerged as a critical need. For example, the AMR FP in Nigeria recommended: “*A dashboard that generates reports and alerts easily*”. Improving the capacity of laboratories to transport samples, detect AMR, and report data was also one of the most common comments from the respondents. For instance, the AMR FP in DRC recommended: “*Equip laboratories with biomedical analyses, develop procedures relating to the transport, collection and analysis of samples*”. Strengthening AMR surveillance systems for the collection of quality data was also a recurring suggestion. The AMR FP in Mali was among the ones that recommended stronger surveillance systems by stating, “*Organize resistance surveillance at the community level and the expansion of sentinel sites for surveillance*”. Other themes from the suggestions include capacity building to improve the collection and handling of AMR data, better financing of AMR research and knowledge-sharing, and creating awareness.

### 2.5. Reducing the Incidence of Infection

The AMR FPs in Guinea Bissau, Equatorial Guinea, Niger, and Togo perceived their countries as neglecting the supervision and assessment of IPC and WASH measures to prevent the emergence of antimicrobial-resistant pathogens in hospitals and communities (Figure 2). Comoros, Eritrea, Liberia, Namibia, Uganda, and Zambia were not conducting surveillance for healthcare-associated infections (HCAIs). Guinea Bissau and Equatorial Guinea were not training healthcare workers on limiting the development and spread of AMR. Cameroon and Chad were not providing Personal Protective Equipment (PPE) and immunization to prevent the spread of antimicrobial-resistant infections. IPC and WASH guidelines in Benin, Cabo Verde, Ghana, Equatorial Guinea, Gambia, Uganda, and Zambia did not incorporate surveillance for antimicrobial-resistant pathogens in communities and the environment (Figure 2).

All the 36 AMR FPs identified the best ways to reduce the incidence and prevalence of AMR in their countries. Half of them highlighted the establishment of IPC measures as a major intervention. Among them was the AMR FP in Kenya, who suggested “*Strengthening IPC practices* e.g., *hand hygiene, vaccination, farm biosecurity*”. Education and awareness to reduce the incidence and occurrence of infections were also recommended by the AMR FP in Cabo Verde, who highlighted “*Education and awareness*” as an appropriate intervention. Surveillance and monitoring were also recommended by several AMR FPs, including Cameroon’s AMR FP, who said, “*xpand IPC, HCAI surveillance coverage to majority of health facilities*”. Other common suggestions include establishing antimicrobial stewardship (AMS) and HCAI programs, training and equipping healthcare workers to manage patients with infectious diseases, and establishing regulations on the use of antimicrobials to reduce the incidence and prevalence of AMR.

### 2.6. Optimizing the Use of Antimicrobials

AMR FPs in Equatorial Guinea, Mauritius, Mozambique, Chad, and Togo indicated that the healthcare facilities and the ministries of health in their countries lack structures and strategies to support the implementation of AMS interventions (Figure 2). Half of the MS had active hospital-based AMS programs (Figure 4a), but only a few of the programs were standalone (Figure 4c). Angola, Cabo Verde, Comoros, and Guinea Bissau were not enforcing laws and regulations regarding the quality control, distribution, and use of antimicrobial agents (Figure 2). Investment in research and development to address AMR was nonexistent in eight countries, a little in eighteen countries, and moderate in nine countries (Table 1). Effective rapid, low-cost diagnostic tools were not integrated for optimizing the use of antimicrobials in the various sectors in Angola, Cameroon, Comoros, Guinea Bissau, Equatorial Guinea, Mozambique, Namibia, Chad, Gambia, and Uganda (Figure 2). Twelve countries have integrated the WHO’s Access, Watch, and Reserve (AWaRe) categorization of antibiotics into their National Essential Medicines List (NEML) (Figure 4b). Prescribers and dispensers in Angola, Burundi, Guinea Bissau, Equatorial Guinea, Mauritius, and Mozambique were not adhering to evidence-based guidelines as a key strategy to optimize the use of antimicrobials (Figure 2).

The 36 AMR FPs suggested specific interventions for their countries to optimize the use of antimicrobials. The most common interventions were formulating and enforcing regulations on the use of antimicrobials, establishment and operationalization of AMS programs, and development, revision, or implementation of a national guideline on AMR. DRC’s AMR FP suggested, “*Monitor the acquisition, distribution and use of antimicrobials, raise awareness and provide ongoing education to antimicrobial prescribers and consumers*”.

### 2.7. Promoting Sustainable Investment for NAP on AMR

Twenty of the AMR FPs indicated that cost-benefit analyses were not included in the fight against AMR in their countries (Table 1). Similarly, 26 of the MS lacked a dedicated budget line for AMR. Eleven countries had not developed resource mobilization capacity to support AMR (Table 1). Benin, Cameroon, Comoros, Eritrea, Ghana, Equatorial Guinea, Sierra Leone, Chad, and Zambia had not progressed in building the capacity to develop and use new tools to support the fight against AMR (Figure 2).

The AMR FPs recommended dedicated budgets for NAPs on AMR. For example, the AMR FP in Guinea thought that their country should “*create a budget line in the state budget for the financing of AMR*”. There was also consensus that the government should prioritize the advocacy, planning, and integration of the fight against AMR in other government operations. For instance, Cabo Verde’s AMR FP saw the need to “*get the commitment from the government to prioritize AMR as a national public health issue*”. Another recommendation was the establishment of a TWG and collaborations among partners to streamline activities and fundraising efforts toward the fight against AMR. The AMR FP in Togo suggested the need to “*put in place the AMR TWG and link it to high level governance structures (Prime Ministry, for example, or the head of state)*”. Other thoughts included setting aside funds for monitoring and evaluation, research, and creating awareness about AMR.

## 3. Discussion

### 3.1. State of “One Health” Multisectoral Collaboration and Coordination among Stakeholders

MCCCs are among the main aspects of the regional AMR strategy reported by the AMR FPs in the WHO/AFRO MS. The reported existence of functional AMR MCCCs in 26 member states agrees with the results of the 7th round of the 2023 Tracking AMR Country Self-Assessment Surveys (TrACSSs), which found an AMR MCCC in 22 countries [15,16]. The MS applied one health collaboration and coordination in addressing AMR threats, which can be sustained through legally binding governance mechanisms for a strong, swift, and accountable action to combat AMR [17]. The integration of collaboration and coordination activities with other cognate strategies can be enhanced to optimize the available human and financial resources toward the optimal functioning of the AMR MCCC.

As most of the AMR FPs recommended, awareness campaigns are necessary to enhance coordination and collaboration among policymakers, non-governmental organizations, and communities [9]. The awareness campaigns will enable them to appreciate the need to allocate adequate financial and human resources, establish structures and monitoring systems, and integrate AMR strategies with other cognate strategies and initiatives such as integrated disease surveillance and Universal Health Coverage (UHC). Capacity building on cost and budgeting tools, which WHO/AFRO has already implemented in 15 countries [15], can be leveraged to enhance resource mobilization for AMR activities.

The WHO/AFRO MS without an AMR MCCC need support to establish them. Most of the countries without an AMR MCCC have integrated approaches and joint working activities on AMR [15]; hence, establishing an AMR MCCC will augment the initiated efforts. The countries with AMR MCCCs should improve their functioning. Benin, Burundi, and Namibia particularly need urgent support to operationalize their AMR MCCC.

### 3.2. Level of Preparedness to Raise Awareness and Conduct Education on AMR

Overall, the WHO/AFRO MS were inadequately prepared to raise awareness and conduct education on AMR. The low response rate to the questions about the preparation and poor ratings of the initiatives to target learning institutions with the AMR information and integrate AMR content into the curriculum of higher learning institutions show suboptimal commitment to raising awareness and educating on AMR. The data in the TrACSS also shows that only 15 member states had nationwide initiatives to raise awareness on AMR, and only one MS conducted the campaigns regularly [16]. Yet, knowledge and awareness of AMR are poor across societal strata in WHO/AFRO MS [18]. The accessibility of professional education and training on AMR ought to be increased across the various sectors, including human health, veterinary, and agriculture, to change stakeholders’ behaviors toward embracing One Health activities to combat AMR [19].

Establishing collaboration frameworks among diverse stakeholders, including the government, civil society, media, and the public, is necessary for a coordinated and concerted effort to substantially raise awareness and conduct education on AMR in WHO/AFRO member states. IPC committees with membership from human and animal health facilities can be established and mandated to ensure comprehensive training and awareness of healthcare professionals regarding AMR [20]. Secondly, as Fuller et al. [18] suggested, AMR education should be institutionalized through integration into curriculums, and continuing AMR education materials should be developed following the WHO’s template [20]. Since the AMR content remains scantly integrated into the curriculums of the learning institutions in the WHO/AFRO, the recommendation by Fuller et al. [18] should be prioritized. An e-learning multisectoral AMR surveillance in-service training curriculum developed by the Ministry of Health in Kenya in partnership with collaborators has been widely accepted [21], which shows that governments in the WHO/AFRO MS can initiate education and awareness interventions successfully.

### 3.3. Evidence Generation and Building Knowledge on AMR for Decision Making

WHO/AFRO MS are generally lagging behind in leveraging surveillance systems to generate evidence to support decision-making in the fight against AMR. Similarly, the TrACSS reports that only 16 MS apply antimicrobial use data to inform decision- and policy-making [16]. The finding agrees with Iwu et al. [19], who flagged the AMR surveillance and monitoring systems in the WHO/AFRO as poor. The countries should strengthen their surveillance and research systems as recommended in the GAP on AMR to generate quality information for evidence-based AMR activities [22]. The TrACSS shows that about half of the member states are establishing integrated surveillance systems for AMR [16]; other MSs should follow suit to establish systems for generating more knowledge and evidence to support decision-making regarding AMR.

The recommendation by AMR FPs to establish laboratories with the capacity for AMR surveillance is consistent with the reality that WHO/AFRO MS lack effective integrated laboratories for AMR surveillance [19]. Countries can leverage the genomics infrastructure established during the COVID-19 pandemic to improve their capacity to detect and provide a high-resolution AMR profile of resistant pathogens within a short turn-around time [23]. A dashboard technology can further support the real-time communication of the AMR profiles and associated prevention and mitigation information [23]. The real-time monitoring of AMR can facilitate tailor-making of responses and strategies for timely prevention and control of AMR.

### 3.4. Status of Implementation of Initiatives to Reduce Infections to Address HCAIs

WHO/AFRO MS have barely implemented initiatives such as IPC and WASH measures to tackle HCAIs. The TraCSS data also show that only six MS have nationally implemented IPC programs, plans, and guidelines [16]. WHO/AFRO should continue supporting MS to integrate One Health AMR surveillance systems that collect data on HCAIs in their practice to inform the development of evidence-based interventions to reduce infections [15]. The surveillance systems should not only be established but also collect trustworthy, valid, and useful data. The surveillance data can help the understanding of the interplay between AMR and HCAIs, which can inform the design of evidence-based safety protocols and IPC practices [24].

The increasing enrolment of African countries and submission of AMR data to GLASS should be accompanied by systematic analysis of collected data to generate relevant information for evidence-based practice and policy changes toward reducing the incidence of infections [25]. Thus, technical personnel addressing AMR should be educated on how best to utilize the AMR data to enhance the effectiveness of interventions. As the AMR FPs recommended, countries should strengthen AMR surveillance by improving education and awareness of AMR and enhancing the resilience of One Health surveillance systems for AMR [26].

### 3.5. State of Readiness in Optimizing the Use of Antimicrobials

WHO/AFRO MS have attempted to implement antimicrobial stewardship and promulgate laws and regulations for antimicrobials. The TrACSS survey reported that only two countries have nationally implemented guidelines for the optimal use of antimicrobials and AMS programs [16]. Most MS have huge rooms for improvement in research and development, application of diagnostic tools, and adherence to evidence-based guidelines to optimize the use of antimicrobials. Boltena et al. [27] reported a low rate of evidence-based implementation of antimicrobial treatment guidelines in sub-Saharan Africa at only 45%. Since adherence to the guidelines depends on the existence of diagnostic tools for culture and susceptibility testing, low-income countries with infrastructural deficiencies can quickly adopt emerging advancements in point-of-care testing and laboratory diagnostic strategies that are not culture-intensive to ensure antimicrobials are only used based on verified needs [28].

Despite the relative advances in the presence of laws and regulations compared to other initiatives to optimize the use of antimicrobials in WHO/AFRO MS, the AMR FPs mainly recommended regulations on the use of antimicrobials and optimization of national guidelines on AMR. This could be due to the suboptimal implementation of the prescription-only sale of antibiotics regulations and inadequate integration of WHO AWaRe classification and AMS principles into national clinical guidelines for the management of infectious diseases [18]. Considering the scarcity of standalone AMS programs in WHO/AFRO MS, the existing infrastructures and systems should be leveraged to support the establishment of AMS programs in healthcare settings, including those in low-resource settings [29].

### 3.6. Promotion of Sustainable Investment for NAPs on AMR

The low rate of AMR NAP-related cost-benefit analyses and the lack of dedicated budget lines for the NAPs show the need to prioritize them in government planning. The finding is consistent with an observation by Harant [30] that information about funding allocations for AMR was missing in 14 of the 15 African countries included in their analysis. Most WHO/AFRO MS depend on external funding rather than government allocations to fund the promotion of AMR NAP. Policymakers in Eswatini and South Africa recommended that governments should allocate funds to One Health for the implementation of AMR NAP [31]. MS also need specialized human resources to conduct cost-benefit analyses, budgeting, and resource mobilization for initiatives to promote sustainable investment for NAP toward gaining the political commitment of governments and international partners to fund AMR NAP. The WHO working paper 2.0 recommends formalized secondments and inclusion of AMR activities in the job descriptions of technical staff [32]. 

### 3.7. Implications for Effective Implementation of the Regional AMR Strategy

This study pinpoints WHO/AFRO MS that WHO AMR FPs consider to be lagging behind in implementing various aspects of the regional AMR strategy. Adopting the AMR regional strategy may be challenging for such countries. The findings of this study can inform stakeholders, including the governments of MS and the WHO Regional Office for Africa, on the priority interventions that need more urgent support in specific MS so that none of them derail the achievement of the regional strategy’s 2030 targets.

### 3.8. Limitations

The risk of self-report bias, considering that the AMR FPs subjectively completed the questionnaire, may have reduced the validity of the findings in this study. Secondly, since this study was an online survey, given the necessity of maintaining the anonymity of responses, it was not possible to ask follow-up questions to clarify some of the AMR FPs’ suggestions. Thirdly, the lack of responses from 11 WHO/AFRO MS and the skipping of some questions by some AMR FPs reduced the completeness of the picture of the state of aspects of the AMR strategy in the WHO/AFRO. However, data from the TrACSS and evidence from the literature that were considered when discussing the results mitigated the self-report bias and improved the completeness of the information since our findings were largely consistent with them.

### 3.9. Recommendations

The WHO/AFRO MS that have not established multisectoral “One Health” collaboration and coordination committees on AMR require support through awareness, multisectoral coordination, and leadership capacity-building. Campaigns and nationwide communication programs on One Health AMR collaboration and coordination should be established as recommended in the regional strategy. The preparation to raise awareness and conduct education on AMR can be improved by supporting MS to institutionalize AMR education, integrate AMR concepts into the secondary and tertiary education curriculums, and develop online continuing professional education modules to make AMR education seamlessly accessible. AMR surveillance systems in WHO/AFRO MS can be improved by guiding the countries to use the genomic infrastructure established during the COVID-19 pandemic to generate knowledge and evidence on AMR. They can apply the dashboard technology to generate quality data and make real-time evidence-based recommendations on how to combat AMR.

The commitment to prevent and fight infections, including HCAIs in WHO/AFRO MS, can be increased by supporting the countries to establish One Health AMR surveillance systems that monitor HCAIs alongside AMR. Healthcare professionals can be empowered through awareness and capacity-building initiatives to ensure they optimally utilize the data generated from the One Health Surveillance Systems in IPC and WASH measures. Empowering WHO/AFRO MS to develop standalone integrated AMS programs in healthcare facilities or leveraging the existing IPC/WASH programs can optimize the use of antimicrobials. Governments of WHO/AFRO MS should budget for AMR and allocate resources to implement AMR strategies in line with the regional strategy’s recommendation to support anti-AMR actions and investments with clear cost-benefit analyses. The MS also need support in the capacity building of the professionals involved in AMR activities to optimize their budgeting, cost-benefit analyses, and resource mobilization needed to promote sustainable investments for NAPs.

## 4. Materials and Methods

An online survey was conducted using the cross-sectional study design. Purposive sampling was applied to recruit one WHO AMR FP from each WHO/AFRO MS to answer the survey questions. AMR FPs are key informants on the implementation of the AMR regional strategy since they are experts who actively support AMR implementation in their countries and are members of the national AMR governance and coordination committees.

The study team developed a structured questionnaire with questions aligned to the six priority interventions in the regional AMR strategy. The questionnaire was optimized through repeated reviews by the study team. It focused on obtaining expert opinions on the extent to which MS had implemented AMR activities identified as priority interventions in the regional strategy. Each of the priority interventions was covered by three to seven closed questions in which the AMR FPs rated the implementation of the various strategies under each of the six priority interventions using a 5-point Likert scale (not at all = 1, a little = 2, moderately = 3, quite a bit = 4, extremely = 5). An open-ended question per priority intervention sought suggestions on the approaches to fast-track its implementation.

The AMR unit in the WHO Regional Office for Africa, Brazzaville, Congo, shared a link to the online survey with the AMR FPs in the 47 MS. The data from the survey was extracted into Microsoft Excel for analysis. For the perspectives data quantified using Likert scales, the total scores of the sub-scales were obtained by adding the scores of the items in the sub-scales, then an overall total score was calculated. Percentages of the total subscale scores to the maximum obtainable scores were calculated. The percent scores were categorized into four quarters as follows: inadequate (0–25%), basic (26–50%), intermediate (51–75%), and advanced (76–100%) implementation. The qualitative data from the open-ended questions were analyzed thematically to identify the AMR FP’s suggestions on the ideal approaches to implementing the priority interventions in the WHO/AFRO MS.

## Figures and Tables

**Figure 1 antibiotics-13-00943-f001:**
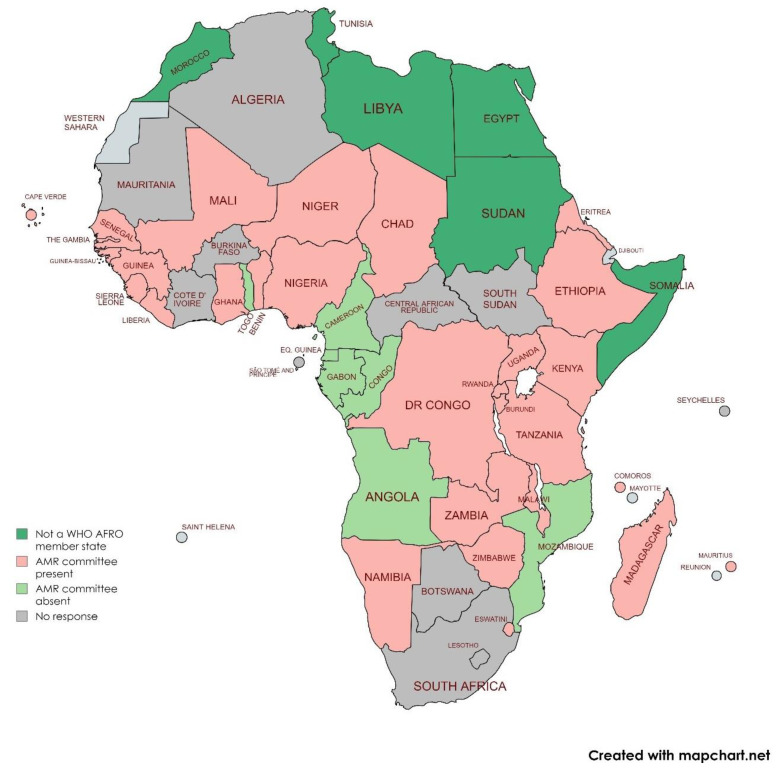
WHO/AFRO member states with an AMR multisectoral collaboration and coordination committee.

**Figure 2 antibiotics-13-00943-f002:**
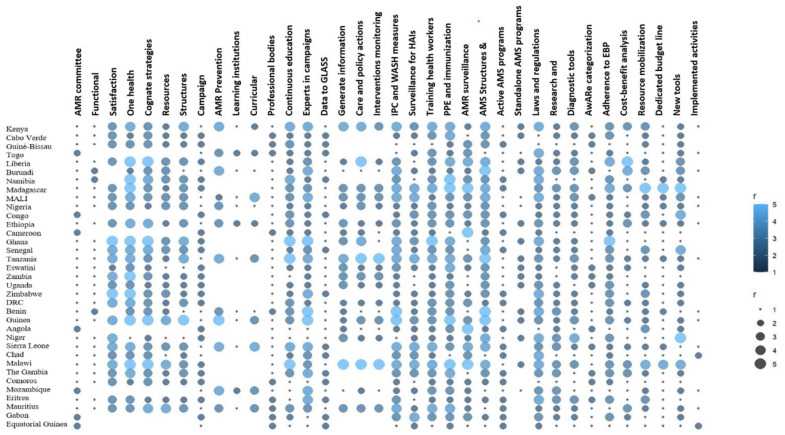
Ratings of the various aspects of the AMR strategy by the WHO AMR focal points.

**Figure 3 antibiotics-13-00943-f003:**
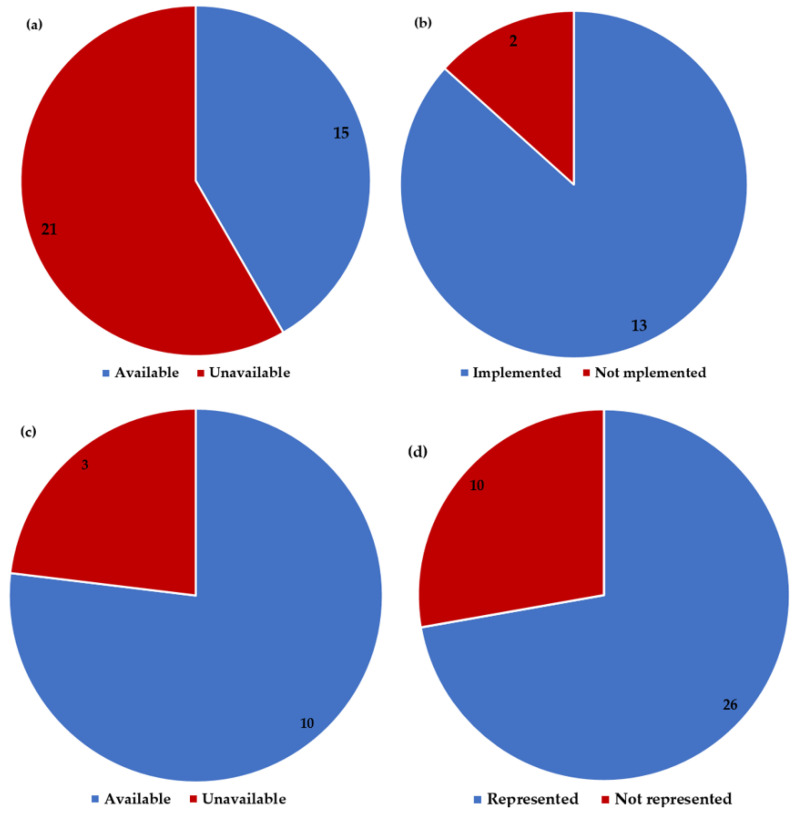
Education and awareness on AMR: (**a**) Proportion of countries with communications and education and awareness campaign strategy on AMR; (**b**) Countries that have implemented the campaign strategy on AMR; (**c**) Availability of educational and awareness campaigns targeting secondary and tertiary learning institutions; (**d**) Representation of professional bodies and institutions of higher learning in the AMR coordination committee and AMR technical working groups.

**Figure 4 antibiotics-13-00943-f004:**
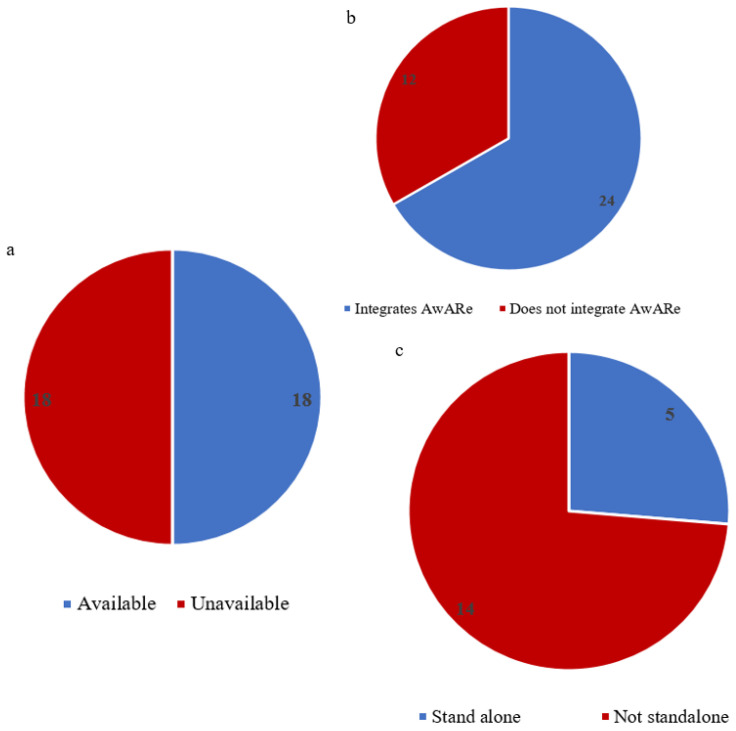
(**a**) Availability of active hospital-based antimicrobial stewardship programs; (**b**) Proportion of countries that have integrated AWaRe categorization of antibiotics into National Essential Medicines list or treatment guidelines; (**c**) Proportion of countries with standalone hospital-based antimicrobial stewardship programs.

**Table 1 antibiotics-13-00943-t001:** Numbers of AMR FPs who rated the various aspects of the AMR Strategy.

Attribute	Responded	Not at All	A Little	Moderately	Quite a Bit	Extremely
Multisectoral coordination and collaboration committee						
Satisfied with functioning of AMR MCCC	26	0	1	15	8	2
One Health in AMR activities	29	1	3	9	8	8
Integration with other cognate strategies	29	0	5	16	4	4
Support with financial and human resources	29	3	13	9	4	0
Structured and monitored implementation	29	4	9	12	3	1
Raising awareness and education on AMR						
Promoting understanding of AMR prevention	13	0	1	5	5	1
Targeting secondary and tertiary learning institutions	13	9	3	0	1	0
Integrated into higher learning institutions’ curricula	13	3	3	5	2	0
Professionals’ continuing education	36	7	12	11	3	3
Integrating experts in planning teams	36	5	12	7	9	3
Building a knowledge and evidence base on AMR						
Generating incidence and prevalence information	22	2	7	9	3	1
Information guides clinical care and policy actions	22	3	7	7	2	3
Information guides monitoring of interventions	22	6	7	7	0	2
Reducing the incidence of infection						
IPC and WASH measures supervised and assessed	36	4	11	9	10	2
Surveillance for hospital-acquired infections	36	7	17	6	6	0
Training healthcare workers to limit AMR	36	2	13	14	6	1
PPE and immunization to prevent AMR	36	2	10	12	8	4
Surveillance for AMR in IPC and WASH guidelines	36	8	12	10	2	4
Optimizing the use of antimicrobials						
Structures and strategies for AMS interventions	36	5	6	13	8	4
Laws and regulations for antimicrobial medicines	36	4	9	10	13	0
Research and development to address AMR	36	8	19	9	0	0
Diagnostic tools to optimize the use of antimicrobials	36	10	16	9	1	0
Adherence to evidence-based guidelines on AMS	36	6	12	14	3	1
Promoting sustainable investment for NAPs on AMR						
Cost-benefit analysis in the fight against AMR	36	20	8	6	1	1
Development of resource mobilization capacity	36	11	15	8	1	1
Dedicated budget line for AMR	35	26	7	1	1	1
Capacity to develop and use new tools against AMR	36	9	19	3	3	2

## Data Availability

The data analyzed in this study can be provided by the WHO Regional Office for Africa upon reasonable request to optimize confidentiality. The qualitative data is from specific national experts whose identities may be potentially deciphered if all the data collected from them is availed in totality.

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
