# Peer review of "Perspectives on the Regional Strategy for Implementation of National Action Plans on Antimicrobial Resistance in the WHO African Region"

_antibiotics, 2024, doi:10.3390/antibiotics13100943_

Round 1

Reviewer 1 Report

Comments and Suggestions for Authors

Some sections need rephrasing for clarity and flow as annotated in the PDF document.

Several abbreviations/acronyms have been used without prior definition e.g. IPC, WASH, AMR.

Ensure that the names of countries are accurately cited for obvious reasons. Examples are annotated in the PDF document.

The quality of the figures can be improved for clarity of text.

Figure 3 is not called out in the narrative. Only Figure 3a is mentioned.

Other comments/suggestions are annotated as appropriate in the attached PDF document.

Comments on the Quality of English Language

The quality of English is great.

Minor rephrasing required for clarity and proper flow, as annotated in the attached PDF document.

Author Response

Dear reviewer, 

Thank you for your comments. Overall, the comments were helpful in improving the clarity of the paper. We have addressed the comments and highlighted the changes in the manuscript for your perusal. Thank you. 

Comment 1: Some sections need rephrasing for clarity and flow as annotated in the PDF document.

Response 1: We have rephrased the sections. "Changes are highlighted throughout the manuscript"

Comment 2: Several abbreviations/acronyms have been used without prior definition e.g. IPC, WASH, AMR.

Response 2: We have provided a prior definition of the abbreviations/acronyms. "Lines 60, 75, 76, 115, 232."

Comment 3: Ensure that the names of countries are accurately cited for obvious reasons. Examples are annotated in the PDF document.

Response 3: We have counter-checked all names of countries to ensure they are accurate. "Lines 133, 148, 176, 182, 184, 231, 252, 256, 261, 262, 283, 284."

Comment 4: The quality of the figures can be improved for clarity of text.

Response 4: We have improved the resolution of the figures for clarity of the labels. "170, 187, 268"

Comment 5: Figure 3 is not called out in the narrative. Only Figure 3a is mentioned.

Response 5: We have called the other parts of Figure 3 in the narrative. "Lines 177, 178, 182, 255, 256, 264

Comment 6: Addresses appear incomplete 

Response 6: We disagree with this comment. We adhered to the requirements of the template when providing the information for the addresses. "Lines 7, 8, 9, 10."

Reviewer 2 Report

Comments and Suggestions for Authors

This paper is focused on a study that surveyed national AMR focal points to assess implementation of six priority interventions, revealing that countries achieved basic-to-intermediate progress, with multisectoral coordination rated highest (62%) and sustainable investment rated lowest (37%).

The introduction is effective in underlining the importance of combating antimicrobial resistance in Africa. However, there is room for improvement, particularly in focusing more on Africa-specific challenges, providing a clearer discussion on the One Health approach, and elaborating on the significance and challenges of National Action Plans. While the global statistics and projections are important, the introduction could benefit from a more balanced approach by shifting more focus towards the specific African context earlier in the introduction. It slso briefly mentions National Action Plans (NAPs) but does not provide enough context or explanation regarding their significance, development, barriers faced by countries in implementing these plans. Since NAPs are central to the study, a more detailed discussion on challenges in the African context would help frame the rest of the paper.

The use of an online survey targeting national AMR focal points provides a structured method of gathering data directly from key informants. The Likert scale-based scoring of AMR priority interventions allows for a quantitative assessment, open-ended questions offer qualitative insights into specific country challenges, however the paper is missing good data analysis. No statistic evaluation of these results is explained in the M&Ms. Only percentage scores are not enough. Likert scales can be difficult to interpret and can be interpreted by parametric tests. It wouldbe good to explain your choices regarding analysis of the information. The open-ended questions provide some qualitative data, but the lack of follow-up questions limits insight. This is important in understanding the nuances of AMR implementation challenges, which might vary significantly between countries and would have been a great strength of the paper.

The results are clearly categorized by the six priority interventions, and show a comprehensive overview of how different countries are performing.

The recommendations provided at the end are practical but they are generic. More country-specific insights and detailed strategies tailored to the social and political environments of different African nations would enhance the impact of these recommendations.

Overall this paper provides good insights into the progress and challenges of AMR strategy implementation in the WHO African region and future studies would benefit from more comprehensive data collection methods.

Author Response

Dear Reviewer, 

Thank you for your comments. We have considered them and edited the manuscript where necessary as highlighted to improve it. However, some of your comments only required clarifications as follows: 

Comment 1: The introduction is effective in underlining the importance of combating antimicrobial resistance in Africa. However, there is room for improvement, particularly in focusing more on Africa-specific challenges, providing a clearer discussion on the One Health approach, and elaborating on the significance and challenges of National Action Plans. While the global statistics and projections are important, the introduction could benefit from a more balanced approach by shifting more focus towards the specific African context earlier in the introduction. It slso briefly mentions National Action Plans (NAPs) but does not provide enough context or explanation regarding their significance, development, barriers faced by countries in implementing these plans. Since NAPs are central to the study, a more detailed discussion on challenges in the African context would help frame the rest of the paper.

Response 1: We structured the introduction using a funnel approach. Only paragraph 1-3 underline the importance of combating AMR globally and specifically in Africa. Only paragraphs one and two provide the global context and effect of AMR. Paragraph three describes the AMR burden and effects in Africa. Paragraphs 4-7 are solely about Africa-specific issues of the implementation of the regional AMR strategy. The WHO Regional Office for Africa already justified its development of the regional strategy in its reports by outlining the challenges of NAP. This study is specifically on the status of implementation of the regional AMR strategy, hence the dedication of most of the introduction to it rather than to NAP that was the basis for the development of the regional AMR strategy. Paragraph four discusses the implementation of one health approach and national action plans in Africa. Paragraph five outlines the Regional AMR Strategy (African region), which WHO intends to use to enhance the implementation of national action plans as indicated in paragraphs six and seven.

Comment 2: The use of an online survey targeting national AMR focal points provides a structured method of gathering data directly from key informants. The Likert scale-based scoring of AMR priority interventions allows for a quantitative assessment, open-ended questions offer qualitative insights into specific country challenges, however the paper is missing good data analysis. No statistic evaluation of these results is explained in the M&Ms. Only percentage scores are not enough. Likert scales can be difficult to interpret and can be interpreted by parametric tests. It wouldbe good to explain your choices regarding analysis of the information.

Response 2: This study is descriptive, only focusing on describing the perspectives of national AMR focal points on the status of the implementation of the regional strategy in their member states. Thus, there was no need for inferential statistical analysis such as the parametric tests that you are proposing. We did not have any question or objective that required estimating association or making causal inferences, hence inferential statistics would not be appropriate for our study objective. We only used Likert scales to facilitate a structured capturing of the intensity of the perspectives of the AMR FPs, hence our data is more qualitative than quantitative and our quantitative data analysis could only be limited to the statistics acceptable in qualitative research, percentages. We have rephrased the sentence on data analysis to clarify so after your comment. "Lines 488 and 489."

Comment 3: The open-ended questions provide some qualitative data, but the lack of follow-up questions limits insight. This is important in understanding the nuances of AMR implementation challenges, which might vary significantly between countries and would have been a great strength of the paper.

Response 3: We have identified the lack of follow-up questions as a limitation of this study, and added a phrase to clarify it further upon your concern. Our efforts to keep the responses anonymous given the sensitivity of providing country-specific data denied us the opportunity to conduct in-depth interviews that would include follow-up questions. Country-based studies can collect in-depth data on the aspects highlighted in this study to obtain the nuances of the implementation of the regional AMR strategy. "Lines 434 and 435."

Comment 4: The recommendations provided at the end are practical but they are generic. More country-specific insights and detailed strategies tailored to the social and political environments of different African nations would enhance the impact of these recommendations.

Response 4: We generalized the recommendations because the AMR FPs cited nearly similar challenges that can be addressed using common strategies. Country-specific studies can be conducted to augment this study by determining the country-specific context and refining the recommendations to suit the contexts.